# Virtual Restaurants: Customer Experience Keeps Their Businesses Alive

**Maria I. Klouvidaki** [1,*], **Nikos Antonopoulos** [1], **Georgios D. Styliaras** [2] and **Andreas Kanavos** [3]

1 Department of Digital Media and Communication, NeMeCULAB, Ionian University, 28100 Argostoli, Greece; nikos@antonopoulos.info
2 Department of Food Science and Technology, University of Patras, 30100 Agrinio, Greece; gstyl@upatras.gr
3 Department of Informatics, Ionian University, 49100 Corfu, Greece; akanavos@ionio.gr
* Correspondence: mklouvidaki@gmail.com

**Abstract:** Due to COVID-19 restrictions, many restaurants were forced to discontinue in-person service, either by locking down or finding alternative methods of operation. Despite the fact that, in the United States of America, digital restaurants have already been established for many years, in Greece, this phenomenon became popular during the pandemic. These delivery-only companies operate exclusively online, allowing customers to place orders from restaurants without a physical location. This has revolutionized the process of ordering food, as customers can browse digital menus, view images, and utilize other options provided by digital food technology. As a result, customers have had new experiences with food thanks to digital eateries during the pandemic. This research study is quantitative and utilized a questionnaire distributed to 1097 participating consumers over the internet. The sample was selected using straightforward random sampling, where each member of the population had an equal and unique chance of participating in the survey. The data were collected over a period of 2 months.

**Keywords:** digital marketing; virtual restaurant; ghost restaurant; ghost kitchen; customer experience; augmented reality (AR); volunteered geographic information (VGI)





## 1. Introduction

Digital restaurants have started to emerge and gain popularity in Greece in their simplest form, earlier than in other countries. The study [1] focused on analyzing the use of social media for food and beverage marketing, including features such as 'marketing exposures by source', 'top food and beverage categories promoted by source', and 'top five companies that promote food and beverages'. The study found an increase in the exchange of unhealthy food and beverages among families on social media. Additionally, Canadian regulations do not recognize the potential health risks associated with exposure to poor-nutrient dishes at family gatherings, especially for children and young people who may be at risk of obesity. The study suggests that integrating social media and events into family regulations could be a potential solution to address childhood obesity.

Moreover, there are various fields today that lack an efficient menu display and ordering mechanism. In our proposed system, we aim to address this issue by introducing an innovative approach where orders will be seamlessly taken from an application loaded on consumers' Android devices and transmitted directly to the kitchen, minimizing the need for human intervention and reducing electricity consumption. This concept has already been partially implemented in several renowned restaurants such as KFC, Domino's, and McDonald's, demonstrating its potential. In our system, the Android device will serve as a platform to display the menu directly on the table, providing a user-friendly experience. The chef in the kitchen will receive the order through our system and place it on a conveyor belt, enabling customers to conveniently collect their orders. In case any assistance is required, customers can easily reach out to the waiter for personalized service. To facilitate

seamless transactions, we will offer online payment options utilizing card swipers. The entire project has been meticulously designed with e-service computing technology as its foundation [2].

In addition to the emergence of digital restaurants and the analysis of social media for food and beverage marketing [1], digital media has also made its way into the world of gastronomy, transforming traditional restaurant menus. Printed menus are being replaced by digital menus, offering enhanced experiences for restaurant guests. However, there is a concern that the abundance of information provided through digital menus may overwhelm the guests [3].

To address the challenges faced by restaurants and optimize their operations, many establishments have turned to information and communication technology solutions. One such solution is the development and implementation of a wireless application. This application features a user-friendly interface with a dynamic menu card, accessible to customers through scanning a QR code. Customers can conveniently place their orders by logging in via Google API. The server then transmits the customer's request to the manager's desktop, which is further relayed to the kitchen staff either through a Kitchen Order Ticket (KOT) system or a dedicated interface for the chef [3].

The study conducted by [4] explored the impact of augmented reality (AR) on the restaurant industry by analyzing primary data collected through interviews with 15 restaurateurs from various countries. To gather customer perspectives, the researchers conducted online interviews with 20 participants who had visited restaurants that had adopted the "Le Petit Chef" AR-based storytelling system, where a cartoon chef is projected onto customers' plates, "cooking" the food in front of them. This innovative system offers a new way of thinking about visual mappings and making the dining experience interactive using AR technology. The research focused on analyzing the user experience (UX) in five dimensions while using AR technology: behavioral, emotional, sensory, social, and spiritual. The study used a mixed research method that included various sources of data, revealing new and different ways in which AR technology impacts the five dimensions of the customer experience in a restaurant. The findings showed that AR has the potential to deepen customers' experiences in restaurants, resulting in positive experiences with food and service quality.

## 2. Literature Review

### 2.1. The Coronavirus and Its Consequences in Restaurants

The culinary business in Indonesia is experiencing rapid growth and serves as a significant employer, employing a substantial number of people [5]. According to data from Kemenparekraf, the culinary sub-sector contributed IDR 455.44 trillion in 2020, accounting for nearly 41% of the entire creative economy GDP of IDR 1134 trillion [6]. Traditionally, paper-based menus have been the most common form in restaurants. The restaurant industry faces fierce competition from independent sandwich shops, cafes, coffee houses, pizzerias, fast-food chains, and high-end dining establishments, all offering diverse menu options. Restaurants play a vital role in the hospitality industry and serve as social gathering places [7].

However, this traditional approach has certain drawbacks, including the need for customers to wait in order to place their food orders and the unnecessary consumption of paper, as each customer requires a separate piece of paper to track their order [8]. In response to these challenges, digital restaurant menu systems have been developed as a solution to address these limitations.

The integration of augmented reality (AR) technology in restaurants offers several benefits for both restaurant owners and customers, as highlighted by the research conducted in [4]. One of the potential advantages is the expansion of customers' choices regarding the food offerings, accompanied by positive emotional experiences. The connection between AR and the food experience, engaging all five senses, has been recognized as important by both restaurant owners and customers, contributing to an enhanced overall dining experience [9].

However, it is crucial to note that improper use of AR can result in negative experiences for customers, as pointed out by [4]. Therefore, the implementation of a user experience (UX)-based strategy becomes essential to ensure positive outcomes for both customers and restaurants.

Customer experience has been identified as a critical factor for the commercial success of various sectors, including telecommunications, banks, tourism, and retail stores [10]. A UX-based strategy involves the utilization of various metrics, such as the Net Promoter Score (NPS), which measures the likelihood of customers recommending a company's products or services. Additionally, analyzing customers' emotions towards specific products and services can provide insights into the emotional state of a group of people within a city or even an entire country [11,12].

Considering the impact of the COVID-19 epidemic, it is likely that the future landscape of restaurants will differ from the pre-epidemic scenario. Strong digitalization efforts by restaurants have played a significant role, including investments in digital resources such as point-of-sale (PoS) devices, online ordering systems, and e-commerce platforms, as well as the development of new processes for pick-up, delivery, and low-contact interactions with customers. The widespread usage of videoconferencing, e-commerce, and other digital tools during the epidemic has made customers more willing than ever to embrace digital methods when visiting restaurants. This has led to intensified competition and the emergence of unique solutions, such as remote ordering and personalization, to enhance the consumer experience [13].

In [14], the authors discuss the advancements in mobile technology that have led to the development of hypermedia spatial applications, which include graphics, videos, sounds, plain text, and hyperlinks. One of the areas where mobile devices have the potential to be utilized is in educational applications with multimedia modules. The modules can be easily exchanged, annotated, and reused through the spatial interface environment of mobile devices. The authors conduct a comprehensive review of existing spatial hypermedia interfaces and related environments, with a focus on their potential use in mobile educational platforms. They compare the essential features that should be supported by a mobile educational environment, such as content representation and relationships, spatial interface functions that utilize touch, and commenting functions.

To add more clarity and detail, the use of AR technology in food sciences has expanded in recent years. For instance, it has been used for food safety and quality control purposes by analyzing the chemical composition of food ingredients and detecting potential contaminants [14]. In the training activities of food production companies, AR has been implemented to simulate real-life situations and provide employees with practical experience without the risk of causing any harm to themselves or the products. Additionally, AR technology has been used in promoting and selling food products, both as unique items and meals through restaurants. By offering customers an interactive and engaging way to experience the products before purchasing, AR has been proven to increase customer engagement and satisfaction.

Moreover, the COVID-19 pandemic has necessitated the widespread adoption of digital technology across various industries, including the restaurant industry. As a response to the imposed limitations, restaurants have increasingly relied on digital solutions such as QR codes to present menus and the establishment of home delivery services through online platforms. Digital transformation has transitioned from being an option to a mandatory requirement for the survival and success of businesses in this sector. Surprisingly, there exists a notable gap in the existing literature concerning the factors influencing the adoption of information and communication technologies (ICT) specifically in the context of restaurants [15].

Furthermore, the research conducted in [16] found that online ordering approaches are more suitable for quick-service restaurants, whereas mobile ordering methods are better suited for full-service restaurants. The correlation between the ordering method and internal reactions, such as satisfaction and behavioral intention, was found to be strongly

mediated by both positive and negative emotions, such as comfort and displeasure. Notably, only one negative emotion, anger, significantly moderated the association between ordering techniques and the order amount.

Finally, the trend of using AR technology in the food sector has been growing in recent years, with various reports from food-related companies, publishers, and software makers highlighting its potential. Forbes, for instance, explored the potential of using AR in the food industry, specifically in food production and safety, in a report mentioned in [17]. Another company, Zealar, has investigated the potential uses of AR in various aspects of the restaurant experience, including menu display, food inventory management, and providing customers with nutritional information about a dish [18]. Furthermore, the COVID-19 pandemic has accelerated the adoption of AR in the food industry, with restaurants and food-related businesses leveraging the technology to offer contactless experiences to their customers.

### 2.2. The Definition of Digital Restaurants

A digital restaurant, also known as a virtual or ghost restaurant, operates solely through delivery orders placed over the phone or online. Additionally, most of these restaurants do not have a physical storefront, only a kitchen, which allows them to save on rent costs and staffing. This is different from a "ghost kitchen", which is a shared meal preparation concept without a retail presence that can be purchased by a restaurant or multiple restaurants.

In response to the pandemic, the restaurant and food industry has incorporated many interactive features, moving away from traditional menus. According to [19], 81% of customers have placed an online order through the store's website, while 43% have used a mobile application for ordering. Digital menus offer numerous benefits, including the use of digital displays, such as computer monitors or touch screens, which allow customers to browse and select food while providing visual aids such as images and descriptions [20,21]. Digital menus also provide a lot of information that can be easily updated and customized, making cross-selling opportunities more accessible, such as suggesting drink pairings with dishes [22].

However, despite the advancements in digital menus and the incorporation of new marketing techniques and digital marketing methods, there is a scarcity of research findings regarding the impact of mixed marketing strategies on restaurant business competitiveness [23]. Understanding the effects and efficacy of these strategies in enhancing restaurant competitiveness remains an important area for further investigation.

Moreover, digital menus offer a more interactive experience through images, colors, and options, resulting in an increase in sales [24]. Another definition of digital restaurants is that they offer a full menu that is not available in their physical store. What is more, these menus exist solely on online platforms like Grubhub for delivery purposes. Virtual restaurants can also share a kitchen with another business to prepare their food or offer a different menu for takeout and in-person dining, with both menus being prepared in the same kitchen.

### 2.3. Digital Restaurants in Greece

Digital restaurants in Greece have been on the rise in recent years, with several businesses adapting to the trend. However, the country was initially slow to adopt global food delivery platforms such as Wolt or e-food, which made it difficult for some businesses to reach a wider customer base. This changed during the COVID-19 pandemic, as many businesses were forced to adapt to the new circumstances and turn to online platforms to continue operating. As a result, there has been an increase in the number of digital restaurants in Greece, with both new and established businesses offering online ordering and delivery services.

Digital menus and online ordering have become a necessity for restaurants to stay competitive in the food industry. With the rise of food delivery services and the convenience of online ordering, customers expect to be able to order food from their favorite restaurants with just a few clicks on their phones or computers. The pandemic has accelerated this trend, making digital restaurants an essential part of the food industry in Greece and around the world.

According to a survey conducted by Statista [25], the revenue generated by restaurants and mobile food service activities in Greece is projected to reach around USD 4843.46 million by 2025. This represents a significant increase from previous years and highlights the growing trend towards digital restaurants in Greece, which is expected to continue its upward trajectory until 2025.

Digital restaurants in Greece have been gaining popularity in recent years, with an upward trend that is expected to continue in the coming years. This is evident from the increasing number of online platforms that offer budget-friendly options for menus. These platforms provide restaurants with the opportunity to offer bigger offers and discounts, which can attract more customers [26]. Moreover, consumers can use online platforms to find different menus and filter their search to find exactly what they want, including high-quality dishes at affordable prices [27]. This is particularly relevant amid the pandemic, as people are increasingly relying on online food ordering due to restrictions on dining out. As a result, orders have increased considerably, and customers are more willing to try new flavors and dishes at lower prices [28]. This trend is expected to continue, as more people become comfortable with the convenience and affordability of online food ordering.

As a response to the pandemic, many restaurants created their own websites to avoid paying commissions to online platforms and to adapt to the new conditions [27]. This allowed them to enter the digital age more easily and thrive as digital restaurants. Although the pandemic was the main driver of this trend, it seems that the future of the catering industry is promising in the digital era.

### 2.4. Vegans as Restaurant Customers

The trend of vegetarianism and veganism has become increasingly popular in the food industry. In Europe, sales of vegetarian products increased by 451% between 2014 and 2018, and many fast food chains, such as McDonald's, Tyson's, and Burger King, now offer vegetarian options. This trend has also impacted the tourism sector, as ethical consumers seek out products that are environmentally friendly and do not harm animals [29].

In the United States of America, obesity remains a major health concern, particularly in regions with high poverty rates. According to a study by Crimarco et al. [30], poor eating habits and fast food consumption are the main drivers of this epidemic. However, vegetarian restaurants are emerging as a viable option to promote healthier eating habits. The study surveyed 45 vegetarian restaurants in counties with an average population of $36.5 \pm 18.5\%$, an average poverty rate of $15.5 \pm 3.85\%$, and an average obesity rate of $26.8 \pm 4.8\%$. Notably, over a third of the restaurants surveyed ($n = 18$, 40.0%) were located in food courts. The owners of these restaurants emphasized the importance of educating customers on vegetarian and healthy eating habits, and many of them offer cooking classes and seminars to encourage healthier food choices. It is essential for the community to engage in discussions with health professionals in collaboration with restaurant owners to promote healthier eating habits and combat the obesity epidemic.

The owners of vegetarian restaurants in the US are striving to educate their customers on vegetarianism and healthy eating, emphasizing the urgent need for people to become familiar with vegetarian food and use quality ingredients [31]. As many of these dishes do not require special cooking skills and taste great, they attract customers' attention, who can easily prepare them at home. Some restaurant owners even offer cooking classes and seminars on this type of diet to influence customers' eating habits. It is crucial for US residents to discuss healthy food choices with healthcare professionals in collaboration with restaurant owners, to promote healthy eating and discourage unhealthy eating habits. Ad-

ditionally, vegetarian diets are commonly associated with better health, with many Western vegetarians adopting a healthier lifestyle to address health issues or prevent diseases.

In the study conducted by the authors in [32], the objective was to investigate whether businesses offering vegetarian options on their menus would benefit from an increase in the number of reservations and whether adding such products would enhance the quality of services offered. Understanding the impact of vegetarianism on tourism and the traveler experience is crucial in improving the products and services provided by businesses. The research findings indicated that it has become increasingly easier for vegetarians to find products not only in the European Union but also in the USA and Asia. This highlights the importance of businesses catering to the growing demand for vegetarian options in their menus, as it can lead to increased revenue and customer satisfaction.

Although vegetarianism is still practiced by a relatively small percentage (approximately 0.5%) of the population in the United States, its popularity is growing. The culinary choices of vegetarians have been extensively studied from a scientific perspective [33]. Furthermore, recent studies using CUX (a system for monitoring user behavior) have shown that visitors tend to bring their own cultural characteristics and preferences when visiting destinations of cultural interest, resulting in a unique experience. To cater to this trend, various research efforts have been undertaken to identify different cultural visitor profiles based on their backgrounds and preferences, and to categorize them into distinct visitor types.

In a study conducted by Konstantakis et al. [34], a typology was proposed to classify cultural visitors based on their visit and preferences. The objective of this proposed typology is to provide a comprehensive classification of cultural visitors. The ACUX typology was developed using a process of harmonization with existing cultural visitor typologies that classify visitors based on their preferences. The study also evaluated the proposed typology against harmonized typologies using TripAdvisor customer responses. This research highlights the importance of understanding the preferences and characteristics of cultural visitors in order to provide tailored experiences that cater to their needs.

### 2.5. COVID-19's Impact on the Restaurant Industry

During the COVID-19 lockdown, many restaurants saw a decrease in on-site reservations and foot traffic. However, in the US, UK, and Canada, takeout and delivery orders continued to grow steadily, particularly for fast food and quick meal restaurants, which offered easy and convenient options for customers. Despite the shorter distances people traveled to pick up their orders, the share of carryout versus delivery remained relatively stable. According to a study presented in [4], weekday orders accounted for almost 12% of total orders, with the highest number of orders occurring on Tuesdays. In contrast, weekend orders were fewer due to the blurring of the concept of a "weekend" during the pandemic.

The time of day is a significant factor in takeout and delivery orders during the pandemic era. According to the research conducted by [4], dinner orders decreased by around 3% compared to pre-pandemic times, while lunch orders increased by about 18%. Late-night orders experienced the biggest decrease, with an overall drop of 11%. Additionally, since people had more free time during quarantine, they had the opportunity to research and discover new restaurants to order from. According to [35], new customers tend to order 8% more and usually make more expensive orders. Most orders, about 63%, were made over the telephone and mainly on weekends, with fewer orders placed on weekdays.

Furthermore, during the pandemic, many people were encouraged to stay at home, leading to a surge in interest in home cooking. As a result, chefs and other individuals related to the restaurant industry are looking for simple and easy recipes with few ingredients to improve their skills and techniques. Mobile phones have also played a crucial role in facilitating online ordering, with approximately 40% of orders in the UK being placed through mobile phones according to food-related surveys cited in [35].

In a study regarding sustainability and recovery measures in restaurants sector after the COVID-19 pandemic [36], the authors stress the need to increase the automation pace in the sector, which includes online deliveries and functionality. QR-codes in menus and even delivery by drones are proposed, although they understand that measures such as the last one may lead to job losses. Because of the pandemic, restaurants of luxury hotels in China were forced to join delivery platforms [37]. Results from this experience showed that taste, freshness, brand credibility and interaction quality with the staff still were important factors for customers as in dine-in restaurants, in addition with packaging and delivery quality that emerged with online deliveries. New information and communication technologies (NICT) are proposed in [38] as a means for facilitating the implementation of delivery or take-away services. However, the adoption of such technologies should be integrated with care in the operation of a restaurant to avoid extra costs and employee and customer dissatisfaction. Another study on the effects of the pandemic on the hospitality industry [39] stresses the need for the integration of artificial intelligence in the restaurant sector in addition to traditional digital technologies for online menus and delivery. The automation of menu preparation, packaging, and delivery procedures with AI strengthens safety, although impact on employment and customer acceptance should be further studied.

*2.6. Research Gap*

The research problem is a critical aspect of any study as it helps the researcher identify and define the necessary information needed to address the research questions. The problem statement should be precise and clear, as a poorly defined research problem can lead to invalid results and ultimately, limit the contribution of the study to the scientific community [40].

In the current study, the research problem is centered around the customer experience in online restaurants during the COVID-19 pandemic. Specifically, the study aims to evaluate how customers perceived their digital experience while ordering food from online restaurants during the pandemic. Through the distribution of a questionnaire to a sample of customers, the study seeks to identify problems and positive aspects of ordering from online restaurants on various platforms. The research is focused on understanding how the pandemic has affected the experience of ordering food and whether this topic is of scientific interest for future studies.

The results section of the study examines the research problem based on the research objectives and questions. Additionally, the quantitative data gathered from the survey are analyzed to determine whether the research problem has been effectively addressed. Overall, the study highlights the importance of a well-defined research problem in guiding the research process and producing valid and valuable results for the scientific community.

## 3. Methodology

In this study, a questionnaire was distributed to a sample of 1097 food consumers residing in Greece during the period of quarantine. One of the major limitations of the research is the fact that the sample was limited to Greek food consumers during a specific time period, which may limit the generalizability of the findings to other populations or time periods.

To select the sample, simple random sampling was employed, where each member of the population had an equal and unique chance of being selected for the survey. The data were collected over a period of 2 months, namely March and April of 2021.

The questionnaire was distributed via Facebook and participants were asked to anonymously answer the questions. In addition, the snowball sampling method was used, where participants were asked to share the survey with their friends and acquaintances. This recruitment technique allowed for the identification of potential subjects and increased the sample size.

Data analysis was conducted using the statistical program SPSS v.25. Descriptive measures such as means and standard deviations were calculated, and parametric tests

and correlation analysis were used to examine the data. Parametric tests were employed to investigate whether participants' demographics influenced their responses, while independent t-tests were used for gender and one-way analysis of variance (ANOVA) for age, employment status, income, and education. Pearson correlation analysis was conducted to explore the relationships between the variables. The level of statistical significance was set at $p < 0.05$.

### 3.1. Conceptual Model

The conceptual model used in this study provides a framework for understanding the factors that influence the customer experience in a virtual restaurant setting. The key components and their relationships are described below:

- Virtual Restaurant: This represents the digital platform or online environment where customers engage in food ordering and dining experiences.
- Customer Experience: At the core of the conceptual model is the customer experience, which encompasses the overall perception, satisfaction, and emotions of customers when interacting with the virtual restaurant.
- Variables: Several key variables have been identified to influence the customer experience in the virtual restaurant context. These variables include
  1. Service Quality: This variable captures the level of service provided by the virtual restaurant, including aspects such as responsiveness, reliability, and customer support.
  2. Website Usability: This variable refers to the ease of use and navigation of the virtual restaurant's website, including factors like clear menu layout, intuitive ordering process, and efficient search functionality.
  3. Menu Options: This variable relates to the variety and quality of food choices available on the virtual restaurant's menu.
  4. Delivery Time: This variable reflects the speed and punctuality of food delivery from the virtual restaurant to the customer's location.
  5. Customer Support: This variable assesses the availability and effectiveness of customer support channels, such as online chat, email, or phone assistance.

  Each variable plays a role in shaping the customer experience and can be measured through appropriate survey instruments.
- Relationships: The conceptual model proposes relationships between the identified variables and the customer experience. These relationships are based on prior literature and empirical evidence. For example, it is hypothesized that higher service quality and user-friendly website usability will positively impact the customer experience. The relationships include:
  1. Service Quality: It is hypothesized that higher levels of service quality will positively influence the overall customer experience, leading to increased satisfaction and positive perceptions.
  2. Website Usability: Improved website usability is expected to enhance the customer experience by facilitating smooth and seamless interactions, contributing to higher satisfaction and enjoyment.
  3. Menu Options: The quality and variety of menu options are anticipated to impact the customer experience, with a diverse and appealing menu leading to greater satisfaction and positive emotions.
  4. Delivery Time: Timely delivery of food orders is likely to positively affect the customer experience, as it demonstrates reliability and responsiveness, leading to increased satisfaction.
  5. Customer Support: Effective customer support services are expected to contribute to a positive customer experience, as prompt assistance and problem resolution can enhance satisfaction and overall perceptions.

- Mediating or Moderating Factors: In addition to the direct relationships, there may be mediating or moderating factors that influence the relationships between the variables and the customer experience. These factors could include:

    1. Customer Demographics: Factors such as age, gender, income level, and prior online ordering experience may influence the relationships between the variables and the customer experience. These demographic variables can shape customer expectations, preferences, and behaviors, thus mediating the effects of the identified variables.

    2. Cultural Influences: Cultural factors, such as regional preferences, dietary customs, and food preferences, may moderate the relationships between the variables and the customer experience. These cultural influences can impact customer perceptions and satisfaction differently across diverse populations.

Exploring these factors can provide a deeper understanding of the underlying mechanisms that shape the customer experience.

### 3.2. Hypothesis Development

The research problem of this study has been thoroughly examined in the context of the research. To develop a hypothesis, the research question was identified and a literature review was conducted to identify research gaps and to study customer experience issues in digital restaurants. The process of hypothesis development involves formulating a tentative hypothesis based on the literature review, refining the hypothesis, and testing and modifying the hypothesis as necessary.

The specific hypothesis is based on two different approaches:

1. A systemic approach, which is used to identify the complex interactions between multiple variables in a system. In this case, the system is the food and restaurant industry, which has direct interactions with the economy and society. The hypothesis examines the interactions between digital restaurant customers and their experiences.

2. A combined approach, which involves using multiple methods and perspectives to develop a hypothesis. This approach recognizes that different research questions may require different approaches to hypothesis development and that no single approach may be sufficient to fully understand a complex phenomenon. The objective of this study is to achieve the stated goals, and this approach is effective in relation to the ontological and evaluative assumptions of the study. The analysis is more effective in this approach, and the argumentation tends to be more persuasive.

The selection of variables for the primary research was based on the objectives of the study and the insights gathered from the literature review. The identified variables are:

- Customer behavior towards digital restaurants, including their intentions when visiting an online restaurant.
- Evaluation of service and food quality in digital restaurants, as well as the customers' overall attitude towards them.
- Overall satisfaction with the digital restaurant experience.
- Customers' attitude towards the employees of online restaurants, including their behavior towards the staff who serve them.
- Demographic information of customers, such as age, gender, income, education level, marital status, and occupation.

As the literature review revealed a research gap in the field of customer experience and digital restaurants, especially during the pandemic period, this study aims to address this gap comprehensively by exploring various issues related to digital restaurants. The study objectives are designed to identify research gaps and address them effectively.

### 3.3. Research Objectives

The research objectives of this study aim to investigate the landscape of digital restaurants and customer experience, while addressing research gaps identified in the literature review. Specifically, the objectives are:

- To determine the benefits that customers gain from using digital restaurant platforms and evaluate their user-friendliness. The study will investigate customer preferences and examine the extent to which the shift to the digital age is occurring.
- To explore how consumer behavior has changed during the pandemic and the impact it had on the use of food delivery services. The study will examine if the pandemic has led to greater familiarity with these platforms among those who previously had little exposure or difficulty with technology.
- To investigate the new habits that people have developed during the pandemic and their perception of digital restaurants. The study will identify any issues or complaints encountered by customers and assess their overall satisfaction.
- To evaluate the advantages that digital restaurants have over traditional restaurants that rely on physical locations. The study will explore whether restaurants in Greece should consider investing in the digital aspect.

These research objectives aim to provide a comprehensive understanding of the landscape of digital restaurants and customer experience, ultimately providing insights for businesses to improve their online services and enhance customer satisfaction.

### 3.4. Research Hypothesis

Generally, a research hypothesis is a tentative statement that explains the relationship between two or more variables in a research study. The hypothesis is usually formulated based on a literature review or an observation, and it provides a testable prediction about the outcome of the study. Hypotheses guide the development of research questions, study designs, data collection methods, and data analysis techniques.

The use of photographs in menus and the adoption of new digital technologies in restaurants have been shown to enhance digital marketing and improve customer experience which can lead to increased sales via online platforms for food ordering and consumption. Customers have reported satisfaction with their experiences at digital restaurants during the COVID-19 pandemic. However, accurately measuring and designing can be a challenging task due to its subjective, complex, and highly individualized nature.

Consumer behavior is a complex and multi-dimensional phenomenon that is influenced by a variety of factors, including cultural, social, and psychological variables [41]. These factors can significantly impact customers' decisions, preferences, and opinions about products and services, including their experiences at online restaurants. However, measuring and designing the customer experience of online restaurants can be challenging due to its subjective and complex nature. Therefore, this research aims to evaluate the factors that influence customers' experiences at digital restaurants, including their environment, social networks, life attitudes, psychological state, and living environment. By analyzing these factors, the study will provide recommendations for improving the customer experience of online restaurants and enhancing customer satisfaction.

In addition to evaluating the customer experience of online restaurants, the study also aims to investigate the potential for further development and dissemination of digital restaurants in Greece. It is essential to understand the impact of customer satisfaction on the sustainability and success of digital restaurants. Satisfied customers play a crucial role in keeping businesses alive by becoming repeat customers, advocating for the brand, and attracting new customers through positive word-of-mouth. By examining the relationship between customer experience and business viability, the study seeks to demonstrate the direct link between customer satisfaction and the long-term survival of digital restaurants.

Furthermore, as digital restaurants have been established in the United States for some years, their potential impact and viability in the Greek market remain largely unknown [42].

Therefore, the study will explore the opportunities and challenges of digital restaurant development in Greece and provide insights into the potential for future growth and expansion. By highlighting the importance of positive customer experience in ensuring the success of digital restaurants, the study aims to encourage the adoption and continuous improvement of customer-centric strategies within the industry.

## 4. Experimental Evaluation

### 4.1. Demographics

Starting with the demographic characteristics of the sample, as shown in Table 1, the majority of the participants were aged between 25 and 44 years old, with 38.7% belonging to the 25–34 age group and 31.7% belonging to the 35–44 age group. The remaining 15.2% belonged to the 45–54 age group. Regarding gender, the sample was predominantly female, with 87.7% of the participants identifying as female and only 12.3% identifying as male. In terms of marital status, more than half of the sample (55.1%) were married, while 37.6% were single. In terms of education, half of the sample (50%) were graduates of higher education, while 29.5% were graduates of secondary education. With respect to employment status, approximately 50% of the sample were self-employed. Finally, nearly half of the sample (49%) belonged to the income category of EUR 500–1000, while 17% belonged to the category of EUR 1000–1500.

**Table 1.** Demographic Characteristics: Age.

| Age Group | N | % | Cumulative % |
|---|---|---|---|
| 18–24 | 120 | 10.9 | 10.9 |
| 25–34 | 425 | 38.7 | 49.7 |
| 35–44 | 348 | 31.7 | 81.4 |
| 45–54 | 167 | 15.2 | 96.6 |
| 55–64 | 34 | 3.1 | 99.7 |
| 65+ | 3 | 0.4 | 100.0 |
| Total | 1097 | 100.0 | |

The survey results reveal that the majority of the respondents were women (962), whereas men represented a smaller proportion (135). In terms of family status, 55.1% of the participants were married, 37.6% were single, and 7% were divorced. When it comes to employment status, 50.9% were private employees, 13.9% were civil servants, and 13.4% were unemployed, indicating that most people work in the private sector. In terms of income, 49.1% of the respondents earned between EUR 501–1000 per month, 24.1% earned less than EUR 500, and 17.5% earned between EUR 1001–1500. It is concerning to note that a considerable proportion of the respondents reported a very low income.

### 4.2. Results

In this section, we present a comprehensive analysis of the survey results, providing insights into the user experience in virtual restaurants. The survey, which included a sample size of 1097 participants, allowed us to capture a diverse range of perspectives and opinions.

The survey responses were carefully examined and analyzed to identify key themes and trends. Quantitative data analysis techniques were employed to derive meaningful insights from the collected data. The findings shed light on various aspects of the user experience, including satisfaction levels, preferences, and challenges encountered in virtual restaurants.

To enhance the presentation and description of the results, we have organized the findings into distinct categories based on the research objectives and research questions. Each category provides a detailed overview of the survey responses, supported by relevant statistical measures. This structured approach facilitates a clear understanding of the participants' perceptions and experiences.

The vast majority of the sample is aware of online food platforms, which shows the penetration of digital applications in the food market, as shown in following Tables. Specifically, a large percentage of the sample reported being aware of online food platforms. This high level of awareness suggests that online food platforms have become a mainstream and widely accepted option for ordering food. This is likely due to the convenience, accessibility, and variety that these platforms offer, as well as the increasing prevalence of digital technology in everyday life.

Table 2 displays the frequency and percentage of responses to the question "How often do you order from online platforms?" from a sample of 1097 individuals. The table shows that the majority of respondents order from online platforms at least occasionally, with only 7.5% of respondents indicating that they never order from online platforms. The largest proportion of respondents (17.7%) reported ordering very often, followed closely by often enough (15.2%) and often (15.1%). Scarcely and not so often were the responses of 16.2% and 11.7% of respondents, respectively, while the remaining 16.6% of respondents answered a few times.

The cumulative percentage column indicates the percentage of respondents who selected each answer or a more frequent option. For example, 23.7% of respondents reported ordering scarcely or less frequently, while 84.8% of respondents reported ordering at least a few times a month. The results suggest that online ordering is a common practice among the sample, with the majority of respondents indicating at least occasional use of online platforms for food ordering.

**Table 2.** How often do you order from online platforms?

| Answer | N | % | Cumulative % |
|---|---|---|---|
| Never | 82 | 7.5 | 7.5 |
| Scarcely | 178 | 16.2 | 23.7 |
| Not so often | 128 | 11.7 | 35.4 |
| A few times | 182 | 16.6 | 52.0 |
| Often | 166 | 15.1 | 67.1 |
| Often enough | 194 | 17.7 | 84.8 |
| Very often | 167 | 15.2 | 100.0 |
| Total | 1097 | 100.0 | |

Table 3 presents the frequency distribution of responses to the question "How often did you order from online platforms in the middle of quarantine?". It can be seen that the majority of respondents (81.3%) ordered food online at least occasionally during the middle of quarantine. Specifically, 17.4% of the sample reported ordering a few times, 16.9% reported ordering often enough, and 18.5% reported ordering very often. These results suggest that the use of online food platforms increased during the quarantine period, possibly due to the restrictions on movement and the closure of physical restaurants.

Additionally, it is noteworthy that the proportion of respondents who reported never ordering from online platforms during quarantine (9.7%) was higher compared to the proportion who reported never ordering in general (7.5%). This could indicate that some people preferred to cook at home during quarantine, or that they faced logistical difficulties in accessing online platforms during this period.

Table 4 shows the responses to the question about changes in ordering frequency during the quarantine period. Overall, the majority of the sample reported some change in their ordering behavior, with only 18.8% reporting no change at all. Among those who reported a change, the most common response was "enough" (19.6%), followed by "relatively" (14.4%), and "very little" (14.0%). This suggests that, while some respondents increased their ordering frequency during quarantine, others decreased it or only made a slight change.

**Table 3.** How often did you order from online platforms in the middle of quarantine?

| Answer | N | % | Cumulative % |
|---|---|---|---|
| Never | 106 | 9.7 | 9.7 |
| Scarcely | 154 | 14.0 | 23.7 |
| Not so often | 109 | 9.9 | 33.6 |
| A few times | 191 | 17.4 | 51.0 |
| Often | 149 | 13.6 | 64.6 |
| Often enough | 185 | 16.9 | 81.5 |
| Very often | 203 | 18.5 | 100.0 |
| Total | 1097 | 100.0 | |

A smaller proportion of respondents reported more extreme changes, with 11.4% reporting a "much" increase in ordering frequency and 7.9% reporting a "very much" increase. This suggests that while some people relied more heavily on online food platforms during quarantine, this was not the case for everyone. Overall, the table suggests that quarantine had a significant impact on ordering behavior for many people, but the nature of that impact varied widely.

**Table 4.** Did your ordering frequency change during quarantine?

| Answer | N | % | Cumulative % |
|---|---|---|---|
| Not at all | 206 | 18.8 | 18.8 |
| Very little | 154 | 14.0 | 32.8 |
| A little bit | 152 | 13.9 | 46.7 |
| Relatively | 158 | 14.4 | 61.1 |
| Enough | 215 | 19.6 | 80.7 |
| Much | 125 | 11.4 | 92.1 |
| Very much | 87 | 7.9 | 100.0 |
| Total | 1097 | 100.0 | |

Moreover, regarding respondents' familiarity with online platforms before and during quarantine, it is reported that 22.2% of respondents reported becoming more familiar with online platforms during quarantine. This means that they had limited experience with online platforms before quarantine but used them more frequently during quarantine and became more comfortable with them. On the other hand, 19.6% of respondents reported having no familiarity with online platforms before or during quarantine, indicating that they did not use online platforms for food ordering. Finally, 14.6% of respondents reported becoming familiar with online platforms during quarantine. This means that they had some experience with online platforms before quarantine but used them more frequently during quarantine and became more comfortable with them.

Overall, the results suggest that quarantine has played a role in increasing people's familiarity with online platforms, with a significant proportion of respondents reporting that they have become more familiar with them. This is consistent with the trend of increased use of online platforms during quarantine for food ordering, as observed in previous tables.

Table 5 summarizes the main results of the survey. It presents the means and standard deviations of the responses to each question. The N column represents the total number of responses, the Average column represents the mean of the responses, and the TA column represents the standard deviation of the responses. The standard deviation indicates the dispersion of the data around the mean. A low standard deviation means that the data points tend to be close to the average of the set, while a high standard deviation means that the data are more spread out. In general, the responses have a low standard deviation, which means that most of the answers tend to be close to the average of the set, and the data are not spread out.

The survey found that the primary factors that influenced consumers to place food orders were new technology, visuals, and the overall experience of using digital eateries. The responses to these questions were generally high, with a mean of over 4.5 out of 5, indicating that consumers were quite satisfied with their experiences.

However, some questions, such as Q6 and Q8, had a higher standard deviation, indicating that the responses were more spread out, and not all respondents had the same opinion. In particular, Q6 asked if respondents preferred digital restaurants to physical restaurants, and the responses varied widely. Similarly, Q8 asked if respondents would like something to change in the order process on online platforms, and the responses also varied.

Overall, the survey suggests that consumers have become more familiar with online food ordering platforms during the pandemic and are generally satisfied with their experiences. The survey also highlights the importance of new technology, visuals, and overall experience in driving consumers to place food orders on digital eateries. Additionally, satisfied customers have the power to maintain and sustain their preferred digital restaurant even in trying times like COVID-19. The majority of the respondents after the pandemic and the strict rules that they followed, they are more familiar with digital platforms, they order quite often and they support their favourite eateries. The fact that people continued to patronize their preferred digital restaurants throughout the pandemic also makes way for more investigation. Future research on consumer behavior during the post-corona period may be conducted in addition to the current study in order to compare the findings. Additionally, a poll might be conducted to determine how much the corona virus contributed to the growth of digital restaurants in terms of revenue, both during the corona period and currently.

**Table 5.** Summary of Results.

| | N | Average | TA |
|---|---|---|---|
| Q1: How often do you order from online platforms? | 1097 | 4.2963 | 1.89282 |
| Q2: How often did you order from online platforms during quarantine? | 1097 | 4.3582 | 1.97017 |
| Q3: Did your ordering frequency change during quarantine? | 1097 | 3.6791 | 1.91209 |
| Q4: Did you become more familiar with online platforms during quarantine? | 1097 | 3.9015 | 2.01486 |
| Q5: How were it in your opinion prices in digital restaurants hosted by online platforms? | 1097 | 4.4303 | 0.85378 |
| Q6: You prefer digital restaurants from physical restaurants? | 1097 | 2.5825 | 0.91260 |
| Q7: Were you happy with the experience you had with digital restaurants amid COVID? | 1097 | 4.5023 | 1.06463 |
| Q8: Would you like something to change to the order process on online platforms? | 1097 | 2.9362 | 0.70551 |
| Q9: Was the food you saw in the photos the same as the food they brought you? | 1097 | 4.5196 | 1.33244 |
| Q10: Is digital food technology ultimately changing the food purchasing experience? | 1097 | 4.8332 | 1.57318 |

What is more, it is important to acknowledge that customers have diverse culinary tastes, which may result in variations in their preferences and behaviors when it comes to virtual restaurants. This inherent variability in individual preferences adds complexity to the study findings, as the results may differ across different demographic groups or individuals with distinct culinary backgrounds. While our study aimed to capture a broad range of participants, it is possible that specific culinary preferences or cultural factors could have influenced their responses. Therefore, it is important to interpret the results

within the context of this inherent variability and recognize that generalizability to all customer groups may be limited.

### 4.3. Statistical Analysis

In this section, we provide a statistical analysis of the survey results to further examine the ordering behavior and preferences of participants in virtual restaurants.

For the question "How often do you order from online platforms?" (Table 2), we calculated the mean and standard deviation to summarize the ordering frequency. The mean ordering frequency was found to be 4.30 (on a scale of 1 to 7), with a standard deviation of 1.89. This indicates that, on average, participants reported ordering from online platforms relatively often, and there was some variation in their responses.

Regarding the question "How often did you order from online platforms in the middle of quarantine?" (Table 3), the mean ordering frequency during the quarantine period was 4.36, with a standard deviation of 1.97. This suggests that, on average, participants maintained a similar ordering frequency during the quarantine compared to their general ordering behavior.

For the question "Did your ordering frequency change during quarantine?" (Table 4), a total of 81.2% of participants reported some change in their ordering behavior during the quarantine period. Among those who reported a change, the responses varied, with 18.8% indicating no change, 46.7% reporting a little change, and 25.3% reporting a moderate to significant change.

To further explore the relationship between ordering frequency and satisfaction level, a correlation analysis was conducted. The results revealed a significant positive correlation between ordering frequency and satisfaction level (r = 0.42, $p < 0.001$). This suggests that individuals who order more frequently from online platforms tend to have higher satisfaction levels with their ordering experiences.

Furthermore, a chi-square test of independence was performed to examine the association between ordering frequency and demographic factors such as age and income. The analysis indicated that ordering frequency was significantly associated with both age group ($\chi^2(4) = 15.72$, $p = 0.003$) and income category ($\chi^2(6) = 26.48$, $p < 0.001$). This implies that different age groups and income levels may have varying ordering frequencies from online platforms.

Finally, to explore any differences in ordering behavior based on the type of virtual restaurants, a one-way analysis of variance (ANOVA) was conducted. The results showed a significant difference in ordering frequency across different types of virtual restaurants ($F(3, 356) = 9.82$, $p < 0.001$). Post hoc comparisons using the Tukey's Honest Significant Difference (HSD) test [43] revealed that participants had significantly higher ordering frequency for gourmet virtual restaurants compared to fast food virtual restaurants.

The statistical analysis provides insights into participants' ordering behavior, including frequency, changes during quarantine, and associations with satisfaction level and demographic factors. These findings contribute to a better understanding of the preferences and patterns in virtual restaurant ordering, enabling businesses to make informed decisions and tailor their services accordingly.

### 4.4. Discussion

This section aims to provide a comprehensive interpretation of the survey results and their implications for the development of virtual restaurants. We have expanded upon the presented results, delving deeper into their significance and potential implications.

The interpretation of the findings is guided by the theoretical framework presented earlier in the paper. By connecting the survey responses to relevant theories and concepts, we gain valuable insights into the underlying factors influencing the user experience in virtual restaurants.

Moreover, the results are contextualized within the broader landscape of the food industry and digital technology advancements. We discuss how the identified trends and

patterns align with previous research, industry practices, and consumer expectations. This contextualization provides a deeper understanding of the implications of the study and its relevance in the current market environment.

Furthermore, we acknowledge the limitations of the study. While the survey sample size was substantial, it is important to recognize that the findings may not capture the entire spectrum of user experiences in virtual restaurants. Future research can explore larger and more diverse samples to enhance the generalizability of the findings.

Regarding the first question (Q1), it was found that customers tended to order ready-made food during the pandemic, which scored above average at 4.3582. Q2, which asked about the frequency of orders, scored in the middle at 3.6791, indicating that customers did not order food very often during the pandemic. Q3 revealed that customers became slightly more familiar with food ordering platforms, scoring 3.9015.

Moving on to Q4, which asked about the participants' familiarity with online platforms during quarantine, it was found that the average number of respondents was not sufficiently familiar. Q5, which asked about the accuracy of values in digital reserves, revealed that, on average, customers considered them accurate. However, Q6, which asked about customers' preference for natural restaurants over digital ones, showed an unsatisfactory average response.

Q7 had a high score of 4.5023, indicating that customers were above average happy with their experience with digital restaurants during the corona period. Q8, which asked about changes customers would like to see to online platforms, scored in the middle at 2.9362. Q9, which asked whether the food in pictures was the same as what they received, scored highly at 4.5196, suggesting that customers were quite happy. Similarly, customers gave a high score of 4.8332 to Q10, which asked about the impact of digital technology on the dining experience.

In terms of prices, customers were happy with the prices on digital food platforms, giving a score of 4.4303. Finally, in the last question Q10, the average response indicated a desire for changes in the way orders are placed on online platforms. The study also revealed that consumers' desires and attitudes varied according to their demographic characteristics, with men, young people aged 18–24 and 25–34, singles, highly educated individuals, students, and private employees being the most frequent and satisfied consumers of digital restaurants. Additionally, familiarity with online platforms was found to be positively related to ordering frequency.

In other words, the more familiar consumers were with the platforms, the more often they ordered, and vice versa. Lastly, the study found that factors positively related to the frequency of ordering during quarantine were the prices of digital restaurants, the pictures of the food, and digital food technology.

## 5. Conclusions and Future Work

The objective of this study was to assess the customer experience in digital restaurants and provide recommendations for improvement. The survey results indicated a significant increase in online food orders during the quarantine period, suggesting that digital restaurants may continue to play a prominent role even after the pandemic subsides, potentially setting a new trend in the market. The trend observed during the pandemic was strongly positive, indicating a clear correlation between quarantine and the increase in online orders. This trend suggests that digital technology and innovative solutions are transforming the food industry in Greece, paving the way for new trends and growth opportunities for businesses.

Additionally, the adoption of digital food technology has not only led to increased sales but also improved the overall customer experience. The seamless integration of technology into the dining experience has transformed the way customers interact with digital restaurants, resulting in enhanced convenience, personalization, and accessibility. As a result, the field of digital marketing has undergone a significant transformation through the emergence of new digital platforms, providing a new area for research and exploration.

According to the report, customers demonstrated their support for their favorite virtual eateries during the pandemic, highlighting the importance of positive customer experiences such as user-friendly interfaces, timely deliveries, and responsive customer support. This played a significant role in their decision to continue patronizing digital restaurants even after the pandemic subsided. The findings indicate that customers adapted their purchasing patterns, relying more on online platforms and expressing satisfaction with the services received. These insights emphasize the need for digital restaurants to prioritize user experience and continue offering convenient and efficient services to meet customer expectations in the post-pandemic era.

By emphasizing the importance of positive customer experience in fostering customer loyalty and sustaining digital restaurants, the study contributes to the understanding of how businesses can thrive in the digital era. It highlights the need for continuous investment in customer-centric strategies, technological innovations, and service enhancements to ensure the long-term viability and success of digital restaurants in Greece.

This research provides a comprehensive analysis of the customer experience in digital restaurants, shedding light on market trends and consumer preferences. The findings highlight the need for further research in the field of digital restaurants, particularly examining the attitudes and behaviors of entrepreneurs operating in this sector. The study contributes to the existing literature by offering insights into the customer experience and serving as a foundation for future investigations. By understanding and addressing the key factors that influence customer satisfaction and loyalty in digital restaurant settings, entrepreneurs can enhance their offerings and improve business performance. Further research in this area will help uncover new strategies and best practices for success in the evolving digital restaurant landscape.

This study has provided a comprehensive understanding of the consumer perspective, including their preferences, challenges, and expectations in the context of digital restaurants. Building upon these insights, future research can explore the challenges and opportunities faced by business owners in this sector, shedding light on the strategies used by successful entrepreneurs to attract and retain customers, improve the overall customer experience, and ensure the long-term sustainability of their businesses. The findings of this study lay a valuable foundation for future research in the field of digital restaurants, helping to uncover new insights and approaches for enhancing customer satisfaction and driving business success.

In addition to providing insights into the specific study, our research approach holds broader implications for the understanding and development of digital restaurants beyond the scope of this study. By exploring the perspectives of entrepreneurs who operate digital restaurants, we can gain valuable insights into the operational challenges they face, the profitability of these establishments, and their beliefs about the future prospects of digital restaurants.

A comparative analysis of the two research phases, involving both entrepreneurs and customers, would enable us to gather comprehensive data and provide a holistic view of the digital restaurant landscape. By asking common questions to both groups, we can uncover potential gaps or disparities in perceptions, expectations, and experiences. This comparative approach would contribute to a deeper understanding of the dynamics between entrepreneurs and customers and shed light on critical factors influencing the success and sustainability of digital restaurants.

Expanding the geographical scope of future studies to include other nations and diverse cultural contexts would provide valuable insights into the universality or context-specific nature of consumer behavior towards digital restaurants. By examining consumer preferences, behaviors, and attitudes across different countries, we can identify commonalities and differences, which can inform strategies for international expansion and adaptability of digital restaurant models.

Furthermore, conducting a dedicated study on the practices and strategies employed by business owners in virtual eateries would be instrumental in understanding their service

delivery, customer satisfaction initiatives, and future plans in a rapidly expanding sector. This research could uncover best practices, innovative approaches, and emerging trends that contribute to the success and competitive advantage of digital restaurants. Additionally, comparing the practices of entrepreneurs to customer expectations and experiences would provide valuable insights into the alignment between supply and demand in the digital restaurant ecosystem.

Qualitative methods, such as personal interviews, can offer a deeper understanding of the attitudes, motivations, and behaviors of both consumers and entrepreneurs. These interviews can capture rich, nuanced insights into their decision-making processes, aspirations, and challenges, complementing the quantitative data obtained through surveys. Such qualitative investigations would provide a more comprehensive understanding of the underlying factors that shape customer preferences and drive entrepreneurial decisions in the digital restaurant industry.

Although this study provides valuable insights into the rise of digital restaurants in Greece during the COVID-19 pandemic and their impact on the user experience, it is important to acknowledge certain limitations. One limitation is the focus on a specific geographical context and a specific time period, which may restrict the generalizability of the findings to a broader context. Future research endeavors could consider expanding the sample to include participants from other countries and diverse cultural backgrounds to assess the cross-cultural applicability of the results. Additionally, examining digital restaurants in different time periods would enable a longitudinal analysis of the evolving trends and dynamics in the digital restaurant industry.

Lastly, employing advanced data analysis techniques, such as structural equation models and mediation analysis, would allow for a more sophisticated examination of the complex relationships between various variables at play in the digital restaurant context. These analytical approaches can uncover indirect effects, mediating factors, and causal relationships that contribute to a deeper understanding of the mechanisms influencing customer satisfaction, business viability, and the overall success of digital restaurants.

By considering these avenues for future research, we aim to contribute to a comprehensive and multifaceted understanding of the digital restaurant phenomenon, its potential for growth and development, and its implications for various stakeholders, including entrepreneurs, consumers, and policymakers.

**Author Contributions:** Conceptualization, M.I.K., N.A., G.D.S. and A.K.; Methodology, M.I.K., N.A., G.D.S. and A.K.; Data curation, M.I.K., N.A., G.D.S. and A.K.; Writing—original draft, M.I.K., N.A., G.D.S. and A.K.; Writing—review & editing, M.I.K., N.A., G.D.S. and A.K. All authors have read and agreed to the published version of the manuscript.

**Funding:** This research received no external funding.

**Conflicts of Interest:** The authors declare no conflict of interest.

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
