# Peer review of "Virtual Restaurants: Customer Experience Keeps Their Businesses Alive"

_information, doi:10.3390/info14070406_

Round 1

Reviewer 1 Report (Previous Reviewer 1)

All my questions were answered satisfactorily

Author Response

Reviewer 2 Report (Previous Reviewer 2)

The review refers to the manuscript that has been re-submitted. I am presenting the results of the verification of the new manuscript in the context of the previous comments and suggestions.

The article presents survey research (n = 1097, which is impressive) on user experience in virtual restaurants.

The authors added a new subsection Conceptual Model which explains the framework of the research.

The authors added a new subsection Statistical Analysis. The results now are better interpreted.

 The authors expanded Discussion; however it should include comparisons of the results obtained with those of other authors in a given topic – with references to literature. This is still missing.

I still claim that some questions should be related to individual restaurants due to the very large variety of restaurants available to customers. Customers have very different culinary tastes, so the results of the survey may vary from group to group.

The introduction is too extensive, and it should be split into an introduction and a research background (literature review). Unfortunately, the article lacks sufficient scientific features. The literature has been improved but still poor – only 33 references. 

Author Response

Reviewer 3 Report (Previous Reviewer 3)

I commend the authors for their thorough revisions and for addressing the specific points I raised in my previous review. Their responses demonstrate a clear understanding of the issues and a sincere effort to enhance the manuscript accordingly.

Round 2

Reviewer 2 Report (Previous Reviewer 2)

I see some improvement. In the future, the authors should highlight all the corrections in the text and answer every suggestion in a separated file in detail - not generally as it is now.

This manuscript is a resubmission of an earlier submission. The following is a list of the peer review reports and author responses from that submission.

Round 1

Reviewer 1 Report

Find here some comments to consider for a better version:

1. In a virtual restaurant, the customer experience is investigated instead of the user experience.

2. At least one conceptual model that represents the main approach is required. Please keep in mind the multiple variables.

3. Make a compelling case for how positive customer experience keeps business alive

4. In the discussion section, provide a broader view of how this approach can be used beyond the specific study.

5. Restructure the conclusion indicating at least the main contribution, what objective(s) has been achieved

Reviewer 2 Report

The article presents a survey research (n = 1097, which is impressive) on user experience in virtual restaurants. Unfortunately, the article has many scientific flaws. The questions included in the survey are not of good scientific quality. Moreover, the results were presented, described, and interpreted in a very limited way. The authors showed only the results (in %) of the survey without presenting any statistical analysis. Some questions should be related to individual restaurants due to the very large variety of restaurants available to customers. Furthermore, customers have very different culinary tastes, so the results of the survey may vary from group to group. Chapter 3, the discussion should include comparisons of the results obtained with those of other authors in a given topic. An introduction that is too extensive should be split into an introduction and a research background (literature review). Unfortunately, the article lacks sufficient scientific features. The literature is also poor – only 27 references. 

Reviewer 3 Report

This research study explores the rise of digital restaurants in Greece during the COVID-19 pandemic and its impact on the user experience, providing insights for future research on the topic. The study sheds light on the growing trend of digital restaurants and their potential for transforming the food industry, providing valuable insights for businesses and researchers exploring the topic. The study methodology is comprehensive and well-structured, but to increase the generalizability of the findings, future research could consider expanding the sample to include participants from other countries and time periods.

It would be helpful to provide more context on the implications of these findings and how they can be used to inform the development of digital food platforms. Additionally, it would be useful to acknowledge any limitations of the study and suggest areas for further research.

To improve the conclusion of the paper, the author can emphasize the need for future research on the attitudes and behaviors of entrepreneurs who operate in the digital restaurant sector. The author can also suggest conducting a comparative analysis of both entrepreneurs and customers, which can provide a unique and insightful perspective on the topic.

Minor editing of English language required
